# A Line Matching Method Based on Multiple Intensity Ordering with Uniformly Spaced Sampling

**DOI:** 10.3390/s20061639

**Published:** 2020-03-15

**Authors:** Jing Xing, Zhenzhong Wei, Guangjun Zhang

**Affiliations:** Key Laboratory of Precision Opto-Mechatronics Technology, Ministry of Education, School of Instrumentation and Optoelectronic Engineering, Beihang University, Beijing 100191, China; sakura-xj@buaa.edu.cn (J.X.); zhenzhongwei@buaa.edu.cn (Z.W.)

**Keywords:** line matching, intensity order, uniformly spaced sampling, low texture

## Abstract

This paper presents a line matching method based on multiple intensity ordering with uniformly spaced sampling. Line segments are extracted from the image pyramid, with the aim of adapting scale changes and addressing fragmentation problem. The neighborhood of line segments was divided into sub-regions adaptively according to intensity order to overcome the difficulty brought by various line lengths. An intensity-based local feature descriptor was introduced by constructing multiple concentric ring-shaped structures. The dimension of the descriptor was reduced significantly by uniformly spaced sampling and dividing sample points into several point sets while improving the discriminability. The performance of the proposed method was tested on public datasets which cover various scenarios and compared with another two well-known line matching algorithms. The experimental results show that our method achieves superior performance dealing with various image deformations, especially scale changes and large illumination changes, and provides much more reliable correspondences.

## 1. Introduction

Feature matching has remained an essential engineering task in image processing and has been widely applied in computer vision, including image registration [1], image-based 3D modelling [2], object recognition [3] and pose estimation [4].

Typical feature matching algorithms usually consist of three steps: feature extraction, feature description and feature correspondence. In the first step, salient and stable features are extracted efficiently. Then the descriptors are constructed to encode the appearance of the neighborhood. In the matching stage, the similarity between the descriptors are measured to evaluate the correspondence. Among the various features used in computer vision, point features have been widely studied [5,6,7,8]. However, point features are likely to fail in low-texture scenes, such as industrial environments and indoor scenes. Because the establishment of the description for point features mainly relies on texture information, the number of the extracted point features may also fall drastically. On the contrary, line features are usually abundant in these man-made scenes, which also provide more geometric and structural information. 

Compared with point feature matching algorithms, line matching methods have their own difficulties. The line segments extracted by line detection methods are far from being practical and physical. In some cases, a line segment in the real world may be divided into several fragments by the existing line segment detector [9,10]. Although some merging strategies are adopted, it is still quite hard to obtain unbroken line segments perfectly. Moreover, the location of line endpoints might not be precisely determined. These characteristics lead to multiple-to-multiple correspondences between two actual line segments from the reference image and test image respectively, which make it more challenging to match than point features. Furthermore, the issue of line feature matching is more complicated than point feature matching when constructing feature descriptors because line segments usually have a variety of lengths. In order to obtain the feature description vector with a fixed dimension, the structure of the feature descriptor must be specially designed. 

Although more researchers have contributed on point features, effective and reliable correspondence between lines has also been studied extensively in recent years. The state-of-the-art line matching algorithms can be roughly classified into two groups: the methods that match individual line segments based on local appearance, and the approaches that match line segments in a group by using geometric and topologic information. Nevertheless, there are some methods consider both local appearance and geometric attributes. 

In 2005, Bay et al. introduced a wide-baseline stereo line matching method [11], considering color profiles, topological structures and epipolar geometry. More line segment matches were added iteratively by applying sidedness constraint. Wang et al. [12] proposed a robust descriptor named the mean–standard deviation line descriptor (MSLD) for line matching based on the mean and standard deviation of intensity gradients from neighborhood appearance. The method used a gradient based method proposed by scale-invariant feature transform(SIFT) [5] to build the description matrix. However, there is no special design to handle scale changes. The description is independent of line length so that the dimension of the description matrix can be fixed. Wang et al. [13] designed a line signature by clustering adjacent line segments into groups based on spatial distances between lines. However, texture information was not employed in the method, and the computation cost was fairly large. Fan et al. [14] introduced planar line–point invariants (LPI) to match line segments. The situation under the conditions of affine transformation and projective transformation was discussed. The algorithm was highly dependent on existing point matching methods to construct reliable invariants between coplanar points and lines. Therefore, constrained by point matching methods, LPI failed to handle poorly textured scenes. In 2013, Zhang and Koch [15] developed their previous work, the line band descriptor (LBD) [16], which also considered the mean and standard deviation of the pixel gradients. The rotation angle between the reference image and the test image was estimated by aggregating the direction of the line segments and several geometric attributes were artificially designed to help generate the candidate matching pairs. The final match is obtained by utilizing the spectral technique. Park and Yoon [17] presented a real-time stereo matching method with line segments. Texture information was expressed by encoding binary streams using nonparametric transformation to obtain real-time performance. However, the algorithm relied on the repeatability of the line segment extraction method to ensure the spatial relations between the lines. López et al. [18] proposed a method to merge the segments which may be the fragments of a single line, and conduct matching in an iterative way by considering structural information from line neighborhoods. However, the structure of the line neighborhood was sensitive to the inaccurate location of line endpoints. Jia et al. [19] proposed a coplanar line–points projective invariant named the characteristic number [20] to undertake the matching. Two line segments were assumed to be coplanar, if the intersection of the two lines was close to one of the endpoints. Then, the intersections of the coplanar line segments are used to build the invariant. Short lines less than 20 pixels were considered in their developed version [21]. Li et al. [22,23] presented a hierarchical method to match lines in groups by establishing the structures, named Line–Junction–Line (LJL), from the Gaussian image pyramids. The unique structure was formed by two adjacent line segments and their intersections, which are assumed to be coplanar in 3D space. A gradient orientation histogram was built to describe the structure, also in a SIFT-like way. They developed their work [24], referred to as a V-junction, aiming at raising the efficiency of the algorithm and introducing a 3D line segment reconstruction method. However, the running time increased sharply if the line segments were crowded in the images.

Our previous work [25] demonstrated the effectiveness of matching line segments in the scale space, and dividing the neighborhood of the line segments into sub-regions based on intensity order to fix the dimension of the line descriptor. The descriptors are constructed by examining the gradient information of the pixels, and the pair-wise geometric attributes of neighboring line segments are considered to generate the final results. This paper extends our previous work, aiming to further improve the robustness of illumination changes and scale changes with the following aspects: 1) the local intensity order is examined to build the local feature description rather than gradient information; 2) uniformly spaced sampling is conducted in a multiple concentric ring shaped structure to reduce the dimension of the descriptor and improve the discriminative ability; 3) the approximate scale between the reference and the test image is estimated; 4) outliers are removed by estimating the fundamental matrix of the image pair from the intersections of the candidate line, matching pairs with the random sample consensus (RANSAC) strategy.

## 2. Methodology

An overview of our line matching method is presented in Figure 1, which consists of the following four main parts: multi-scale line segment detection, intensity-based local descriptor construction, descriptor matching in an image pyramid and fundamental matrix estimation by intersections. More details for each part are discussed in the following sections.

### 2.1. Line Segment Detection from Image Pyramid

In order to acquire the scale robustness in the matching process and overcome the fragmentation problem, the line segments are extracted by line detectors in a multi-scale way, which is based on Gaussian scale-space theory. 

The original reference and test images are down-sampled and Gaussian blurred. For convenience, only one layer is built between two contiguous octaves. Then, a line segment detection algorithm is applied to each octave of the image pyramid to generate line segments. The corresponding lines located in each octave image are matched by inspecting several geometric characteristics. First, we check the parallelism of the line segments located in the same region in the octave images. If the angle between two lines is less than 10 degrees, they are considered as parallel. Then the distances between the endpoints are estimated. The scale factor is applied to the endpoints of the lines in each octave images, in order to evaluate the distances under original scale. Lines are considered corresponding only if the endpoints are close enough. After exploring all the line segments in the image pyramid, the corresponding line segments are assigned to an identical group, as shown in Figure 2. 

In the following line matching process, corresponding line segments in the same group are regarded as the identical line segments matching with other line groups. It is worth mentioning that not every line segment in the original image has its corresponding line in other octave images. A group with only one line segment from the original image could potentially exist.

For the line segment detection algorithm, a real-time line segment detector with edge drawing (EDLines) [9] is used in this paper. It is widely used in computer vision applications in recent years [26,27,28]. EDLines generates line segments very fast with no need for parameter tuning. The Helmholtz principle [29] is used in the detector to make validation with few false positives. As mentioned above, EDLines is applied to the octave images to detect line segments.

### 2.2. Intensity-Based Local Feature Descriptor

#### 2.2.1. Support Region with Local Coordinate System

For a line segment detected from the octave image, the local appearance is evaluated by picking and partitioning its local neighborhood, which is the first consideration when creating a feature descriptor. All the subsequent computation is conducted in this local region. A line support region is established to undertake the computation and a local orientation system is built to obtain rotation invariance. 

The support region is designed to be rectangular and centered on the line segment. As Figure 3 shows, the width of the region is set to be equal to the length L of the center line l, and the height is denoted by h. The local coordinate system is built by investigating the average gradient direction of pixels on the line segment, which is defined as $d_{⊥}$, and the anticlockwise orthogonal direction of $d_{⊥}$ is defined as $d_{L}$. Attributed to the definition of the local orientation system based on gradient direction, the rotation invariance is achieved when constructing a multiple concentric ring-shaped structure. The details are explained in the following section.

#### 2.2.2. Sub-Region Division Based on Intensity Order

In this section, the line support region is further divided into several sub-regions. The division process is quite essential, because line segments with different lengths should be unified to the description vector with a fixed dimension, in other words, the descriptor should be independent of the line length, which relies on a particular design. It can be observed that some of the previous line descriptors choose to carry out the division based on spatial information, such as banded structure [12,15], which is parallel to the line segment, in order to overcome this problem. After this, the statistic approaches are applied to each bands to compute the descriptors. However, such geometrical division strategies may not take full advantage of spatial or intensity information. Inspired by the local intensity order pattern [30], we divide the support region into several irregular sub-regions based on intensity order to improve the distinctiveness of the descriptor.

To start, we introduce some basic symbol definition. $P=\left\{p_{1},p_{2},\ldots ,p_{n}\right\}$ is the set of *n* pixels in a support region, and their intensity is denoted by $I=\left\{I(p_{1}),I(p_{2}),\ldots ,I(p_{n})\right\}$. Firstly, those *n* pixels are sorted in ascending order by their intensity value, and then a new array is formed, which is given by $\hat{P}=\left\{p_{f(1)},p_{f(2)},\ldots ,p_{f(n)}\right\}$. The new intensity order set is described as,
(1)$$\hat{I}=\left\{I\left(p_{f\left(k\right)}\right):I(p_{f(1)})\leq I(p_{f(2)})\leq \ldots \leq I(p_{f(n)}),k\in \left[1,n\right]\right\},$$
where f(1),f(2),…f(n) is one of the permutations of 1,2,…,n. Then those sorted pixels are divided into *m* groups uniformly of the same sizes. In particular, the threshold of each group of sorted pixels is given by
(2)$$T=\left\{I\left(p_{f(k)}\right):f\left(k\right)=k\cdot \left\lfloor n/m\right\rfloor ,k\in \left[1,m\right)\right\}.$$

Thus a pixel p(x) in the support region can be assigned to a particular sub-region S(x;T) based on the threshold. It is described as,
(3)$$S\left(x;T\right)=\{\begin{array}{ccc} k, & & I\left(p_{f\left(k-1\right)}\right)\leq I\left(p\left(x\right)\right)<I\left(p_{f(k)}\right),k=1,2,\ldots ,m-1\\ m, & & I\left(p\left(x\right)\right)>I\left(p_{f\left(m-1\right)}\right) \end{array}$$

As Figure 4b shows, each color represents a specific group of pixels. A support region is adaptively divided into several parts based on intensity order, rather than an artificially designed banded or circular structure. According to the construction principle, it is robust to monotonic illumination changes.

#### 2.2.3. Multiple Local Intensity Order Representation

In this section, multiple local intensity ordering is applied to describe the local neighborhood. Sample points distributed on a circle centered on a pixel are used to derive the description. In this work, a multiple concentric ring-shaped structure is established to improve the discrimination of the local feature description rather than a single circle. In addition, sample points are picked at regular intervals and assigned to different point sets to reduce the dimension of the descriptor to further improve the computational efficiency and improve the discriminative ability at the same time.

Firstly, a pixel *x* in a partitioned sub-region in the local coordinate system is considered. A number of neighborhood points are evenly distributed over a series of concentric circles of radius $R=\left\{R_{j}:j\in \left[1,r\right]\right\}$ centered at *x*. As Figure 5 shows, there are *N* sample points on each circle with the same uniform distribution. The *i*th neighborhood point on the circle of radius $R_{j}$ is denoted as $x_{j}^{i}$, where i∈[1,N], j∈[1,r]. The first point $x_{j}^{1}$ on the circle of radius $R_{j}$ is located along the $d_{L}$ direction of the local coordinates from the center point *x*. Then the remaining neighborhood points are sampled and distributed on the circle in an anticlockwise direction, as shown in Figure 5, and the rotation invariance is obtained. The intensity value of those neighborhood points is obtained by bilinear interpolation, which is denoted by $\left\{I\left(x_{j}^{i}\right):i\in \left[1,N\right],j\in \left[1,r\right]\right\}$. After this, the local feature description on each circle is calculated by sorting the intensity value of the *N* sample points in an ascending order to obtain a permutation, which is formed by their superscripts. Mathematically, an *N*-dimensional vector is mapped to a permutation of {i:i∈[1,N]} based on the order of $\left\{I\left(x^{i}\right):i\in \left[1,N\right]\right\}$. To be more specific, the new local intensity order set is described as
(4)$$\tilde{I}=\left\{I\left(x^{f\left(q\right)}\right):I\left(x^{f\left(1\right)}\right)\leq I\left(x^{f\left(2\right)}\right)\leq \ldots \leq I\left(x^{f\left(N\right)}\right),q\in \left[1,N\right]\right\},$$
where f(1),f(2),…f(N) is one of the permutations of the superscripts 1,2,…,N. Thus, each local intensity value sorting corresponds to an exclusive permutation. 

It is quite clear that the total number of the permutations is *N*! for a single circle. In order to facilitate the constructing descriptors, a one-to-one lookup table was established to match the local intensity ordering $\tilde{I}$ with the corresponding permutation. Accordingly, the local intensity ordering $\tilde{I}$ can be mapped directly to the corresponding integer index value η. The lookup table covers all the possible permutations of length *N*!.

Then, a histogram *H* is created to compute the local feature descriptor, which is described as
(5)$$H=\left\{B_{\eta }:B_{1},B_{2},\ldots B_{N!},\eta \in \left[1,N!\right]\right\}.$$

Each bin $B_{\eta }$ in the histogram corresponds to an integer index value of the lookup table. Obviously, the size of the lookup table depends on the number of neighborhood points *N*, and the dimension of the local feature descriptor, which is computed based on the lookup table, also increases dramatically as *N* increases. Therefore, we picked *N* sample points at regular intervals in different point sets to avoid dimension disaster. As mentioned above, *N* neighborhood points are uniformly distributed on each circle. Those points are then uniformly space sampled and divided into *v* sets with *u* sample points in each point set, as shown in Figure 6a, which means N=u⋅v. As Figure 6b shows separately, multiple point sets are established on the circles of radius $R=\left\{R_{j}:j\in \left[1,r\right]\right\}$. They are represented as
(6)$$\{\begin{array}{c} \left\{x_{1}^{1},x_{1}^{v+1},x_{1}^{2v+1},\ldots ,x_{1}^{N-v+2}\right\}\\ \left\{x_{1}^{2},x_{1}^{v+2},x_{1}^{2v+2},\ldots ,x_{1}^{N-v+1}\right\}\\ \vdots \\ \left\{x_{r}^{1},x_{r}^{v+1},x_{r}^{2v+1},\ldots ,x_{r}^{N-v+2}\right\}\\ \left\{x_{r}^{2},x_{r}^{v+2},x_{r}^{2v+2},\ldots ,x_{r}^{N-v+1}\right\}\\ \vdots \end{array}.$$

In this way, the dimension of the local feature descriptor depends on v⋅u!, which is much lower than *N*!. Subsequent experiment sections show that the dimension of the local feature descriptor is reduced while the discrimination and the robustness is maintained.

#### 2.2.4. The Construction of the Local Feature Descriptor

As stated above, the histogram technique is applied to build the descriptor. In order to further increase the robustness of the descriptor and emphasize the importance of the pixels close to the line segment, a Gaussian weight $g_{d}$ is applied rather than simple voting. It is defined as
(7)$$g_{d}=\exp\left(-\frac{d}{2\sigma^{2}}\right),$$
where *d* is the distance of the current pixel *x* to the line segment *l* and σ=h. In particular, for *u* sample points in point set *v* on the *r*th circle centered at pixel *x* in the sub-region *S*, $g_{d}$ is added to the bin $B_{\eta }$ based on their intensity value sorting results and the histogram is denoted by $H_{S}^{\left(r,\nu \right)}$. Then the whole local feature descriptor *D* is constructed simply by concatenating the histograms together and normalization. It is represented as
(8)$$\{\begin{array}{c} H_{1}=\left[H_{1}^{\left(1,1\right)},\ldots ,H_{1}^{\left(1,\nu \right)},\ldots ,H_{1}^{\left(r,1\right)},\ldots ,H_{1}^{\left(r,\nu \right)}\right]\\ \vdots \\ H_{S}=\left[H_{S}^{\left(1,1\right)},\ldots ,H_{S}^{\left(1,\nu \right)},\ldots ,H_{S}^{\left(r,1\right)},\ldots ,H_{S}^{\left(r,\nu \right)}\right] \end{array},$$
(9)$$D=\left[H_{1},H_{2},\ldots ,H_{S}\right].$$

### 2.3. Descriptor Matching in Image Pyramid

In this section, the similarity between feature descriptors is evaluated and the approximate scale between the image pairs is estimated. To address the scaling and the fragmentation problem, we extract line segments in a multi-scale scheme in the previous section and the local feature descriptors are constructed. In Zhang’s work [15], the rotation angle between the image pairs are estimated by drawing a direction histogram of the line segments. Inspired by this, we also compute a minimum distance histogram to estimate the scale between the image pairs. 

The feature descriptors in the image pyramid with the closest scale return the most similar results. From the above consideration, the similarities between all the local feature descriptors in the image pyramid are calculated. A common way to do this is through the calculation of the Euclidean distances of the feature description vectors. To be specific, for a group of identical line segments in the reference image pyramid and a group of identical line segments in the test image pyramid, the descriptors are denoted as $D_{ref}=\left\{D_{ref}^{s_{0}},D_{ref}^{s_{1}},\ldots ,D_{ref}^{s_{\gamma }}\right\}$ and $D_{test}=\left\{D_{test}^{s_{0}},D_{test}^{s_{1}},\ldots ,D_{test}^{s_{\gamma }}\right\}$, where $s_{0},s_{1},\ldots ,s_{\gamma }$ are the corresponding scale factors. The Euclidean distances between $D_{ref}$ and $D_{test}$ are calculated, which are defined as,
(10)$$dist\left(D_{ref},D_{test}\right)=\left\{dist\left(D_{ref}^{s_{i}},D_{test}^{s_{j}}\right):i\in \left[0,\gamma \right],j\in \left[0,\gamma \right]\right\}$$

Then the minimum values of the distances are computed and accumulated to each bin corresponding to each pair of octave images to construct the distance histogram. Not all the minimum values are taken into account, due to the consideration of the influence of some random factors. We only choose to add up the top λ percent of the total number of the line segments from the octave image with the lowest number of line segments. Accordingly, the scale factor corresponding to the bins with the minimum value is considered to be the approximate scale between the original image pairs. An example is demonstrated in the following experimental section.

After estimating the scale factor, the nearest neighbor distance ratio (NNDR) matching criterion is applied to further select candidate matching pairs from the corresponding octave image pairs. It is defined as the distance from the nearest match divided by the distance from the second nearest match. If the distance ratio is within a certain threshold, it is considered to be a candidate match.

### 2.4. Fundamental Matrix Estimation by Intersections

Inevitably, false matches exist after evaluating local feature descriptor with the NNDR criterion. To be specific, candidate matching pairs obtained by a higher NNDR ratio may provide more correct matches but also more outliers. Therefore, we use the RANSAC strategy to remove outliers by estimating a fundamental matrix between the reference image and test image from candidate matching pairs. 

An eight-point algorithm is used to compute the fundamental matrix [31]. Hence, intersections of every two line segments from candidate matching pairs are calculated. Among them, the intersections located out of the image are discarded. For example, $\left(l_{ref}^{i},l_{test}^{i}\right)$ and $\left(l_{ref}^{j},l_{test}^{j}\right)$ are two pairs of candidate matches, and $q_{ref}^{ij}$ is where the two lines $\left(l_{ref}^{i},l_{ref}^{j}\right)$ intersect. Similarly, $q_{test}^{ij}$ is the intersection of $\left(l_{test}^{i},l_{test}^{j}\right)$. Then $\left(q_{ref}^{ij},q_{test}^{ij}\right)$ is considered to be a pair of candidate intersections spontaneously. If both $q_{ref}^{ij}$ and $q_{test}^{ij}$ are located in the image range, they are sent to estimate the fundamental matrix. After carrying out RANSAC, inlier intersection pairs are obtained, and the corresponding line segments that form those intersections are regarded as the final result.

## 3. Experiments and Discussion

In this section, image matching experiments are conducted to evaluate the performance of the proposed line matching method. Two well-known line matching algorithms are also tested in the same scheme to compare the discriminative ability and the robustness of the methods. They are MSLD, the line segment appearance description method with SIFT-like strategy; and LPI, the algorithm based on line-point invariants. All the line segments in the experiments are extracted by the EDLines detector. The point matches required in LPI are obtained by SIFT from the OpenCV library.

The test image pairs used in this paper are shown in Figure 7 and Figure 8, which cover different kinds of scenarios from a natural scene. All the image pairs are from the public datasets downloaded from the internet and used in literature [14,15,30,32]. Common image transformations are collected as follows: rotation (Figure 7a,b), illumination changes (Figure 7c,d and 8), view point changes (Figure 7e) and scale changes (Figure 7g,h). Occlusion (Figure 7f) and low-texture scenes (Figure 7i,j) are also included. 

### 3.1. Parameters Evaluation

In order for the line matching method to exhibit better performance, several relevant parameters should be carefully selected. They are the number of sub-regions *m*, the number of concentric circles *r*, the radius of the circles $R=\left\{R_{j}:j\in \left[1,r\right]\right\}$, the number of sample point sets *v*, the number of sample points in each point set *u* and the height *h* of the support region. To evaluate the performance of the proposed method under different parameter settings, various image matching experiments are conducted with all the tested parameters listed in Table 1**.** When discussing a parameter, all other parameters are set to the same fixed values. The matching precision and recall are used as the evaluation criterion. However, in order to consider matching precision and recall comprehensively, the F1-Measure is also taken into account. They are defined as follows
(11)$$precision=\frac{the\;number\;of\;correct\;matches}{total\;matches},$$
(12)$$recall=\frac{the\;number\;of\;correct\;matches}{the\;groundtruth\;matches},$$
(13)$$F1-Measure=\frac{2\cdot precision\cdot recall}{precision+recall}.$$

The average performance of the method under different parameter settings are shown in Figure 9 by the recall vs 1-precision curves and the curves of the total number of correct matches. The selected parameter values are also displayed in Table 1**.**

For the height of the line support regions, Figure 9a demonstrates that the result is not monotonically increasing with the parameter. The number of correct matches increases rapidly in the beginning, then the performance trends toward a steady and slow decline. 

Figure 9b shows that as the number of sub-regions increases, the number of correct matches rises to a peak and then falls. As shown in the recall vs 1-precision curves, *m* = 6 is slightly superior to *m* = 4, although it achieves the maximum number of correct matches when *m* = 4. 

As can be seen in Figure 9c, more correct matches are obtained when three or four points are sampled in each point set. It is observed that *v* = 3, *u* = 3 outperforms the other cases from the recall vs 1-precision curves.

Figure 9d indicates that multiple concentric circles have a big advantage over the single ring-shaped structure. To strike a balance between the discriminative ability and the dimension of the local feature descriptor, two circles with radius *R*_1_ = 12, *R*_2_ = 15 are selected. 

### 3.2. The Descriptor Dimension

The dimension of the local feature descriptor and the effective measure to reduce the dimension is discussed in this section.

Suppose that the line support region is divided into *m* sub-regions, and *r* concentric circles are constructed with *v* point set on each circle, each containing *u* sample points. The dimension of the proposed local feature descriptor *D* can then be calculated as dim(D)=m⋅r⋅(v⋅u!). The dimension of the local feature descriptor $D^{\prime }$ with only one point set on each concentric circle with the same number of sample points is $\dim(D^{\prime })=m\cdot r\cdot (v\cdot u)!$ by contrast. As can be seen in Figure 10, the dimension of the local feature descriptor is reduced dramatically by dividing the same number of sample points into multiple point sets rather than using those sample points directly. Nevertheless, it is also important to note that the performance might decrease when the sample points are divided, to a certain extent. As shown in Figure 9d, six sample points are divided into two and three point sets, respectively. Obviously, when each point set contains only two sample points, it becomes a binary pattern, which might reduce the discriminative ability of the method.

### 3.3. The Scale Estimation

As described earlier, the scale factor between the original image pairs is estimated by constructing a minimum distance histogram from the image pyramid. Figure 11 and Figure 12 gives an example of a minimum distance histogram of the image pair shown in Figure 7g. The ground truth scale between the image pair is 3.3 and the estimated approximate scale is $2\sqrt{2}\approx 2.83$.

Figure 11a illustrates the image pyramid of the reference image and the scale factor between each adjacent octave image is $\sqrt{2}$. In this case, the octave image in scale four has the least number of line segments, with 49 line segments. So the sum of the top *λ* percent of 49 minimum Euclidean distances are computed and drawn in Figure 12. As can be observed, the summation in scale $2\sqrt{2}$ is the smallest over all the presented values of λ. Hence, it is selected as the approximate scale between the image pairs. As for parameter λ, we select λ=80 in the following experiments.

### 3.4. Experimental Results and Discussion

In this section, we present the details of the line matching experimental results to evaluate the proposed method compared with MSLD and LPI. The same criterion described above is followed and the precision, recall and F1-measure values are used to evaluate the methods. 

#### 3.4.1. Natural Scene Image Pairs

The experiments are conducted on 10 pairs of natural scene images shown in Figure 7. The bar graphs in Figure 13 shows the performance of the methods and Figure 14 illustrates the matching results of our method. 

As can be seen from Figure 13, the proposed method performs better than other two algorithms for image pairs with rotation changes (Figure 14a,b). This is because all the calculations are based on the local coordinate system built by examining the average gradient on either side of the line segment, so that the local feature descriptors are less affected by the rotation changes. In addition, the sampling strategy when constructing descriptors achieves more robustness against rotation changes than dominant orientation estimation system in point features, compared with LPI based on SIFT descriptor. 

For image pairs with drastic illumination changes in Figure 14c,d, the proposed method demonstrates the best performance, mainly due to the design of intensity value sorting. The results indicate that the intensity order-based method is more robust against large illumination changes than dominant orientation estimation technique. 

Image pairs with viewpoint change and occlusion are displayed in Figure 14e,f, and the matching results show that the proposed method obtains high scores in both accuracy and recall. 

In the case of scale changes, illustrated in Figure 14g,h, our method outperforms the other two algorithms significantly. It is observed that the multi-scale line segment detection and matching strategy help to overcome the scale change problem effectively. The SIFT descriptor used in LPI also employs the scale-space theory to handle scale changes. In contrast, as shown in Figure 13, MSLD fails in this scenario because of its lack of special design for scale variation. 

Figure 14i,j shows low-texture scene, which are quite common in indoor scenes. Our method ranks the top of three methods in terms of both precision and recall and obtains fairly high scores. In contrast, LPI with SIFT fails in this situation, because the local appearance described by SIFT descriptor is largely depended on texture information. It is quite difficult to manage such a poorly textured scene with a lack of gradient information. 

#### 3.4.2. The Illumination Dataset

The illumination dataset contains three sets of image sequences with increasing illumination changes, as shown in Figure 8. The performance of the methods is shown in Table 2.

As can be seen from Table 2, the proposed method produces a much greater number of correct matches with the highest precision in all the image sequences. The number of correct matches obtained by our method is twice the amount of MSLD. The proposed method still provides 121 correct matches with high precision under the large illumination changes in Figure 8a(1–5), while the performance of both MSLD and LPI decreases rapidly.

The running time is also evaluated in this experiment. The experiments are conducted on a 1.8 GHz Intel Core i7-8565U processor with 16 GB of RAM. The average running time for each line segment is given in Table 2. The processing time of our method depends on the number of the extracted line segments, the dimension of the local feature descriptor and the area of the line support region.

## 4. Conclusions

We proposed a line matching method based on multiple intensity ordering with uniformly spaced sampling, which demonstrated good performance under a variety of scenarios. The main contributions are as follows: 1) uniformly spaced sampling is introduced at multiple concentric ring shaped structures to improve the discriminability; 2) the dimension of the local feature descriptor is also reduced by extracting sample points at regular intervals; 3) the approximate scale between the image pairs is estimated to handle scale change problems; 4) outliers are removed by computing fundamental matrix estimated by the intersections of line segments from candidate matching pairs.

The relevant parameter settings were analyzed in this paper and the dimension of the local feature descriptor was also discussed. The proposed method was compared with another two well-known line matching algorithms under the same test conditions. The experimental results confirm that our method is robust to rotation, scale changes and large illumination changes. The proposed method tends to reach higher precision and recall values in comparison, and a greater number of correct matches were obtained.

The running time of the proposed method was relatively high. Future work will focus on improving the processing efficiency while maintaining the robustness and the discriminative ability.

## Figures and Tables

**Figure 1 sensors-20-01639-f001:**
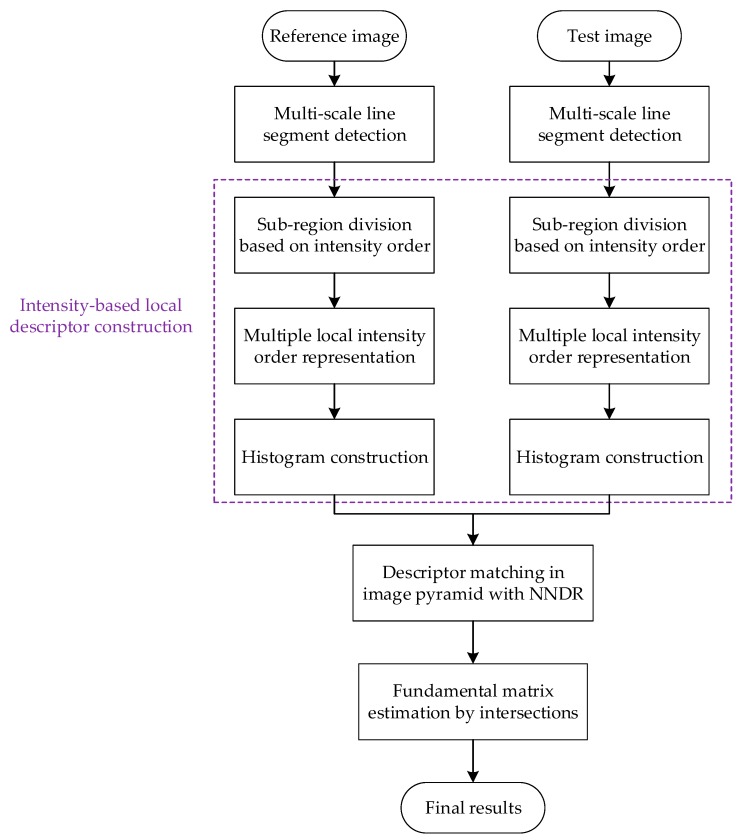
The flowchart of the proposed method.

**Figure 2 sensors-20-01639-f002:**
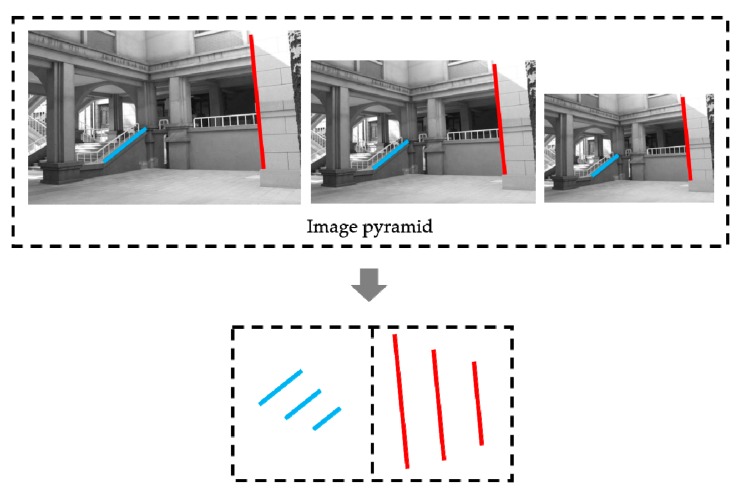
Illustration of the corresponding line segments in the identical group from image pyramid.

**Figure 3 sensors-20-01639-f003:**
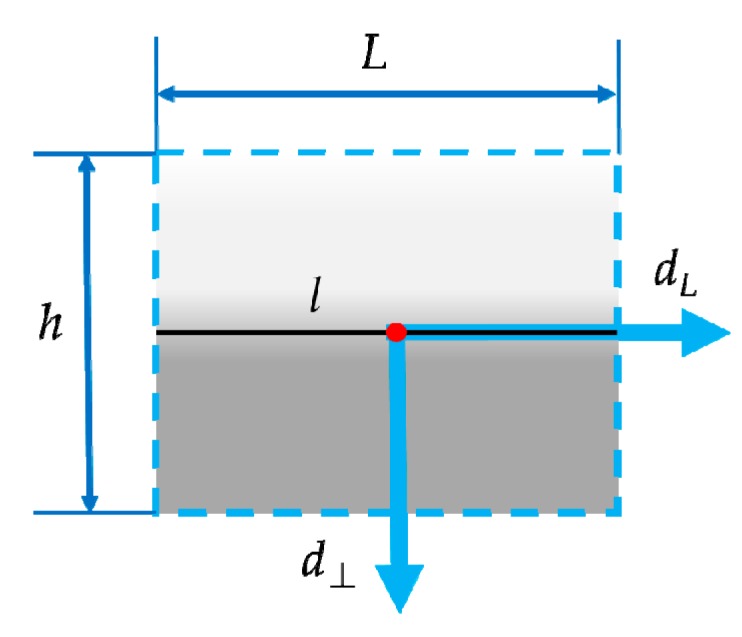
Illustration of the support region with local coordinate system of line segment *l*. The line length is denoted by *L* and the height of the support region is *h*. The direction *d_L_* is defined as the average gradient direction of pixels on the line segment, and its anticlockwise orthogonal direction is denoted by $d_{⊥}$.

**Figure 4 sensors-20-01639-f004:**
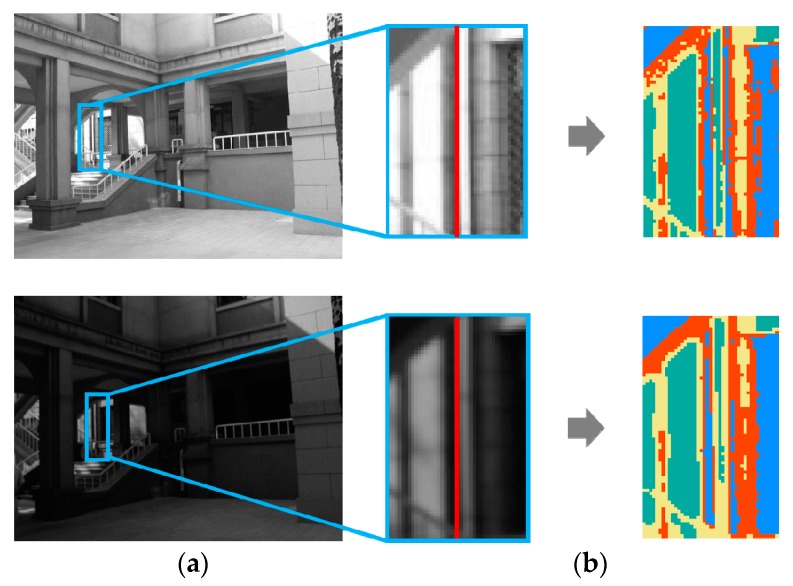
Illustration of adaptively divided support region into multiple sub-regions. (**a**) Local zoom on a pair of corresponding line segments with support region from image pair. The line segments are highlighted in red and the support regions are marked in blue boxes. (**b**) Irregular-shaped sub-regions labeled by different colors.

**Figure 5 sensors-20-01639-f005:**
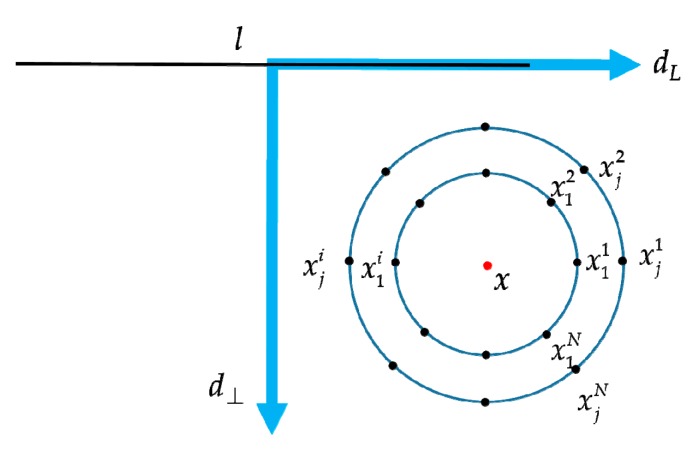
Illustration of the multiple concentric ring-shaped structure based on the local coordinate system with sample points uniformly distributed on the circles centered at pixel *x*.

**Figure 6 sensors-20-01639-f006:**
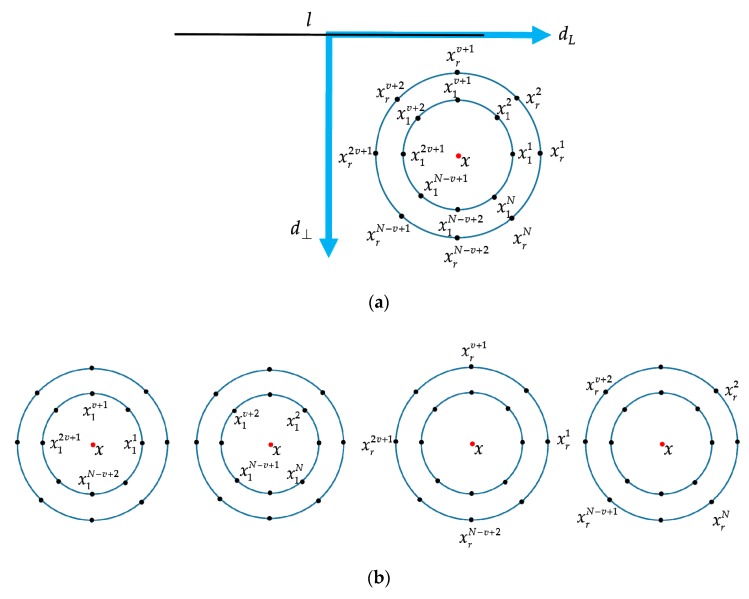
Illustration of the division of the neighborhood points. (**a**) All the neighborhood points of pixel *x* and the calculations are made in the local coordinates; (**b**) *N* neighborhood points are uniformly spaced divided into *v* sets with *u=N/v* sample points in each point set, which corresponds to expression (6).

**Figure 7 sensors-20-01639-f007:**
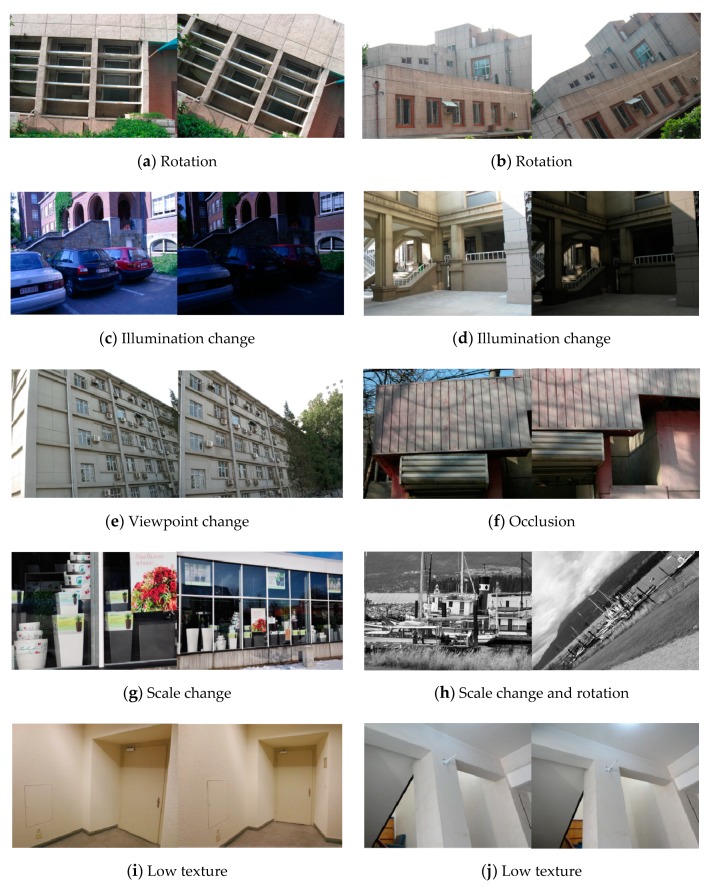
The test image pairs from public datasets of a natural scene.

**Figure 8 sensors-20-01639-f008:**
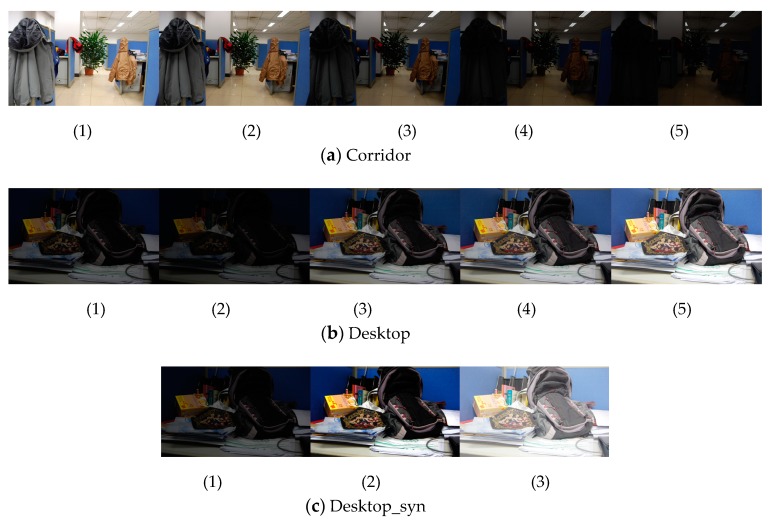
The illumination dataset.

**Figure 9 sensors-20-01639-f009:**
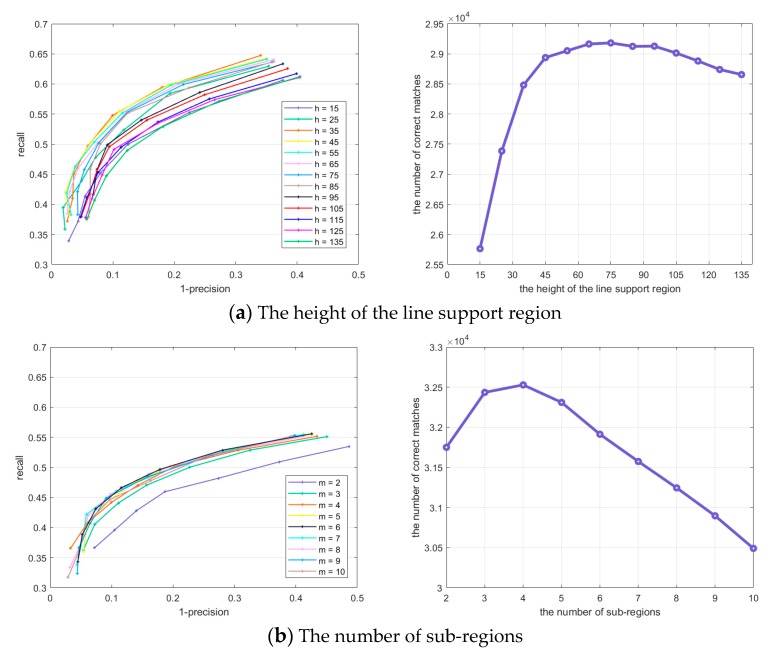
The average performance in terms of recall vs 1-precision curves and the curves of the total number of correct matches under different parameter settings.

**Figure 10 sensors-20-01639-f010:**
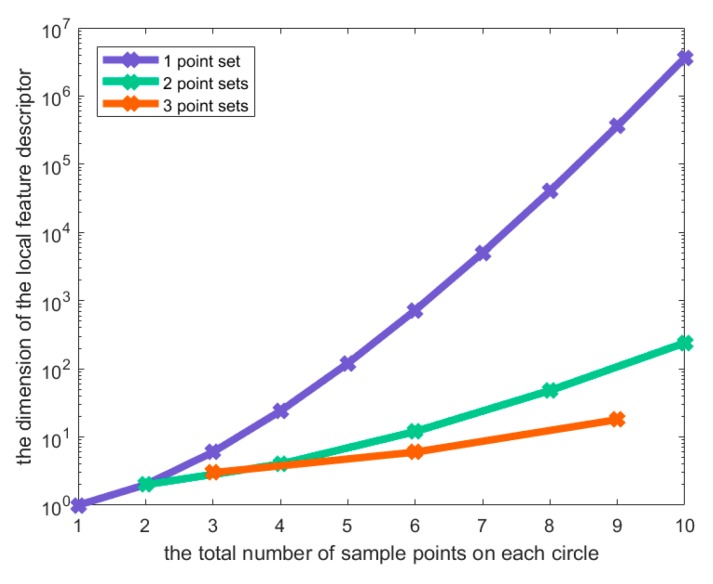
The comparison of descriptor dimensions under different number of point sets.

**Figure 11 sensors-20-01639-f011:**
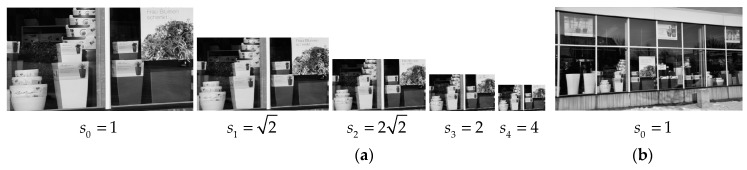
The image pair with scale change. The corresponding scale factors are marked below the images. (**a**) The image pyramid of the reference image with five octave images. (**b**) The original test image.

**Figure 12 sensors-20-01639-f012:**
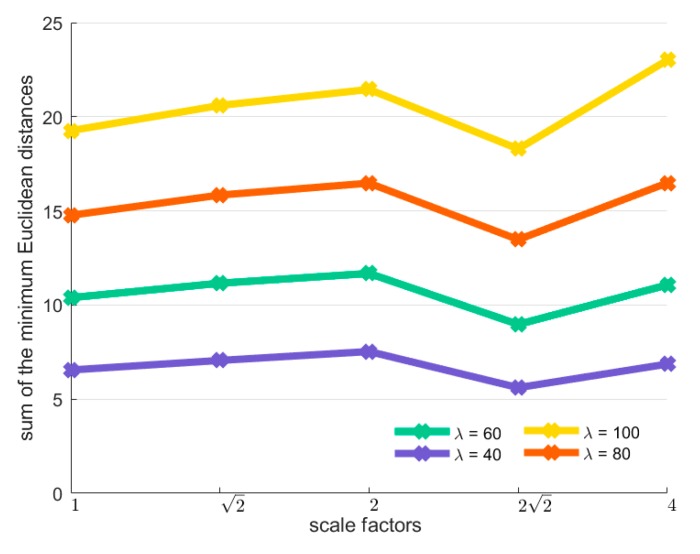
The minimum distance histogram in five scales accumulated by different quantities of minimum Euclidean distances.

**Figure 13 sensors-20-01639-f013:**
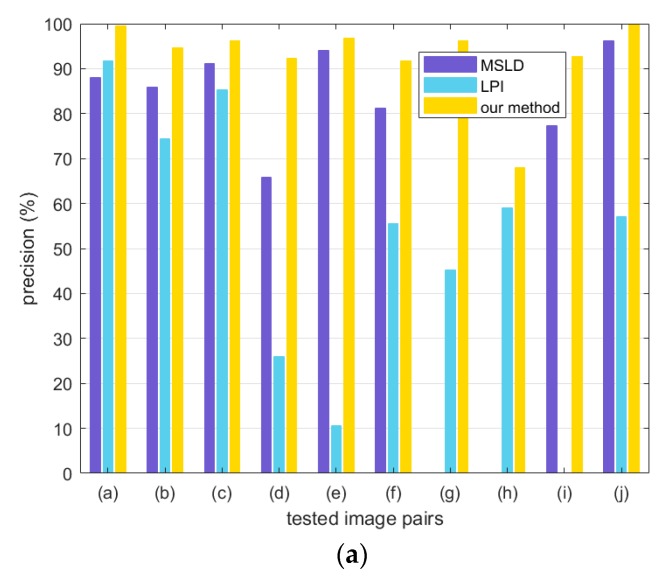
The performance of the mean–standard deviation line descriptor (MSLD), line–point invariants (LPI), and the proposed method. (**a**) precision; (**b**) recall; (**c**) F1-Measure.

**Figure 14 sensors-20-01639-f014:**
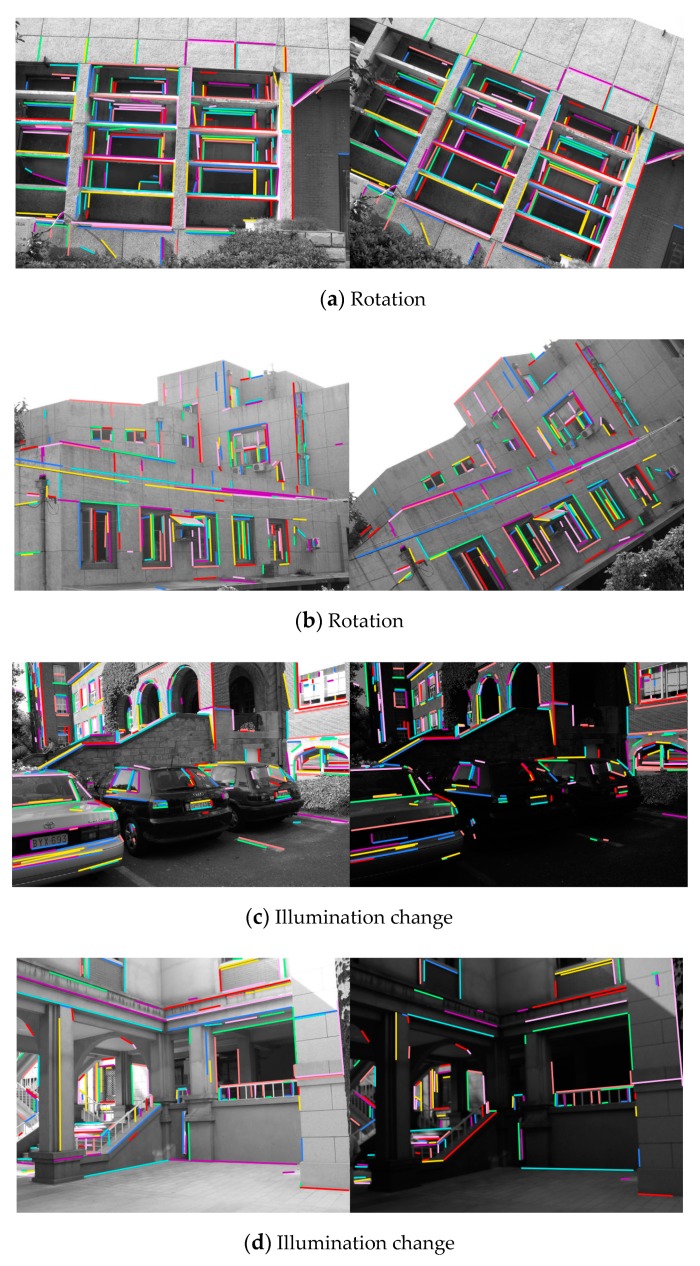
The matching results of the proposed method on different image pairs. Each pair of matched lines is represented by the same color and marked with the same number at the center of the lines.

**Table 1 sensors-20-01639-t001:** Parameter settings determined by experiments.

Symbols	Tested Values	Selected Values	Description
*m*	2, 3, 4, 5, 6, 7, 8, 9, 10	6	the number of sub-regions
*r*	1, 2,3	2	the number of concentric circles
*R*	9, 12, 15, 18, 21, 24, 27, 30	12, 15	the radius of the circles
*v*	1, 2, 3	3	the number of sample point sets
*u*	2, 3, 4, 5, 6	3	the number of sample points in each point set
*h*	15, 25, 35, 45, 55, 65, 75, 85, 95, 105, 115, 125, 135	45	the height of the line support region

**Table 2 sensors-20-01639-t002:** Line matching results of the proposed method, MSLD and LPI on illumination datasets.

Img Pairs	The Number of Img Pairs	Method	Correct Matches	Total Matches	Precision (%)	Average Time (ms)
corridor	1–2	proposed	**763**	775	**98.5**	8.0
		MSLD	376	395	95.2	2.3
		LPI	128	465	27.5	19.1
	1–3	proposed	**453**	466	**97.2**	7.6
		MSLD	202	219	92.2	2.4
		LPI	69	250	27.6	9.2
	1–4	proposed	**243**	257	**94.6**	7.7
		MSLD	96	119	80.7	2.6
		LPI	32	163	19.6	7.3
	1–5	proposed	**121**	127	**95.3**	7.4
		MSLD	29	54	53.7	2.9
		LPI	7	12	58.3	6.0
desktop	1–2	proposed	**299**	307	**97.4**	8.2
		MSLD	179	186	96.2	3.8
		LPI	46	52	88.5	6.7
	1–3	proposed	**509**	515	**98.8**	8.1
		MSLD	302	310	97.4	2.4
		LPI	214	230	93.0	8.3
	1–4	proposed	**491**	495	**99.2**	8.0
		MSLD	281	287	97.9	2.4
		LPI	214	227	94.3	7.6
	1–5	proposed	**371**	375	**98.9**	8.1
		MSLD	239	244	98.0	2.2
		LPI	53	176	30.1	6.0
desktop_syn	1–2	proposed	**411**	414	**99.3**	8.2
		MSLD	276	283	97.5	2.2
		LPI	175	208	84.1	6.9
	1–3	proposed	**361**	365	**98.9**	8.1
		MSLD	181	189	95.8	2.2
		LPI	21	126	16.7	5.2

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
