# Peer review of "A Line Matching Method Based on Multiple Intensity Ordering with Uniformly Spaced Sampling"

_sensors, 2020, doi:10.3390/s20061639_

Round 1

Reviewer 1 Report

The manuscript titled “A line matching method based on multiple intensity ordering with uniformly-spaced sampling” proposed a line matching method based on multiple intensity ordering with uniformly-spaced sampling. And the present experimental results show the proposed method achieved high performance. The line matching method is very significant in matching methods, and the proposed method is novel and work well. Also, the experiments in this paper were well designed, and their results proved high performance of the proposed methods. Therefore, I recommend the manuscript to be published with minor revision. My comments are as followed:

  1. The authors have presented experiment and evaluation methods in the 3rd section “experiment and discussion”, but generally, the experiment and evaluation methods should be given in the method section. I suggest that the authors can consider moving the experiment and evaluation methods description to method section.
  2. English of this manuscript still can be improved. It is better to polish the manuscript by a native speaker.
  3. Line 20. It is better to give the quantitative experiment results in the abstract, which can help readers understand your performance.
  4. Line 90. It is better to give a little detail of your previous study [25], because readers may be not family with this previous study.
  5. The number of each line in Figure 13 is not clearly, especially for Figure 13(e). It is better to zoom out the number or remove the number, because the color has showed the matched lines.

Reviewer 2 Report

The authors propose a new line matching approach using multiple intensity ordering with uniformly-spaced sampling. The work of the paper is interesting.

My comments are listed as the following:

1. The motivation for the research is not very clear. And what are the shortcomings of the existing literature? Why do the authors propose the new approach?
2. L174, "Then those sorted pixels are divided into m groups uniformly. Specifically, the threshold of each group of sorted pixels is given by...", does the "uniformly" here mean that the m groups have the same size? If so, is it possible to divide the sorted pixels into m groups of different sizes by clustering? Another question is what the threshold refers to?
3. Is the feature descriptor rotational invariant? If so, how does it work?
4. The experiments were conducted on only 8 pairs of pictures. The author should compare the proposed approach with MSLD and LPI in larger-scale testing images. In addition, the source of the testing images should be given.
5. What are the results obtained by MSLD and LPI in Fig. 13? In L414-L422, "MSLD fails ... ", "While LPI with SIFT fails in this situation...", how can we get these results from Fig.13?
6. Time complexity is important for image processing and computer vision. I suggest the authors report the running time of the proposed approach.

Reviewer 3 Report

This work presents a line matching detection approach based on multiple intensity ordering with uniformly-spaced sampling. The paper is well-written with good English, but it presents significant drawbacks.

1. First of all is contribution to the state-of-the-art: the manuscript presents a work whose basis is a previous work [25]. The authors claim that their contribution in this new work is the application of a feature descriptor based on local intensities, so it is also robust to rotation invariances (which one is also computed from a previous work [26] - local intensity order map). It is not completely clear the value of those contributions, since the academical core at a first glance is a combination of existing methods.

2: is the methodological aspects for validation: comparison of results was conducted against MSLD and LPI. Since line detection is now more an open problem at all, several approaches were proposed since the traditional Hough line detector. Hough line itself is carried of a set of parameters which in my opinion is a great disadvantage to any computational approach. By varing those parameters (even randomly) some arbitrary combination eventually may achieve a very good result. In other words, is the same we can for the proposed approach: several parameters. If it can be justified, a more detailed comment regarding the execution parameters is needed.  Apparently the proposed methods is very sensible to parameters chance, providing the most varied output results. To be fair, an exploratory parameter execution range experiment must be performed, ranging all possible execution parameters for the proposed approach, and the compared methods. The evaluation index would choose the best overall for that image, for each algorithm.

3: number of algorithm / approaches for comparison must be increased.

4: results presents quantitative evaluation marginally better results when compared to other 2 methods. Nonetheless, qualitatively speaking, several other lines can be observed on tested images, where MSLD, LPI and the proposed approach were unable to properly detect.

5: How much is the time-complexity of the proposed approach in terms of asymptotic analysis (e.g.: Big O) ?  If the proposed approach is able to run in real-time for instance video processing (or nearly), please, provide clear evidences.

Reviewer 4 Report

Page 2 line 58 – the MSLD acronym should be explained

Page 2 line 92 – the LIOP acronym should be explained

Page 5 line 167 – “To start with, some basic symbol definition is introduced. Let P={p1, p2,…, pn}   be the n pixels”. I propose replace with “…be the set of n pixels”

Conclusions:

 what is the impact of research findings on theory and practice in considered field?

What are disadvantages of proposed approach?

The future works should be indicated.

Round 2

Reviewer 2 Report

The authors answered my questions very well and the paper has been improved greatly. I think that it can be accepted for publication.

Two minor errors:

(1) "Error! Reference source not found.." occurs many times in the PDF file.

(2) L484-L486: "is depend on"=>depends on?